# Molecular Mechanisms and Clinical Phenotypes of *GJB2* Missense Variants

**DOI:** 10.3390/biology12040505

**Published:** 2023-03-27

**Authors:** Lu Mao, Yueqiang Wang, Lei An, Beiping Zeng, Yanyan Wang, Dmitrij Frishman, Mengli Liu, Yanyu Chen, Wenxue Tang, Hongen Xu

**Affiliations:** 1Precision Medicine Center, Academy of Medical Science, Zhengzhou University, Zhengzhou 450052, China; 2Basecare Medical Device Co., Ltd., Suzhou 215000, China; 3Translational Medicine Center, Huaihe Hospital of Henan University, Kaifeng 475000, China; 4The Research and Application Center of Precision Medicine, The Second Affiliated Hospital of Zhengzhou University, Zhengzhou 450014, China; 5Wissenschaftszentrum Weihenstephan, Technische Universitaet Muenchen, Am Staudengarten 2, 85354 Freising, Germany

**Keywords:** hereditary hearing loss, *GJB2*, missense variants, molecular mechanisms, deep mutational scanning

## Abstract

**Simple Summary:**

Hearing loss is the most common sensory impairment in humans. Globally, *GJB2* is the most common responsible gene, with missense variants being the most frequent variant type. Pathogenic missense variants in the *GJB2* gene cause isolated hearing loss and hearing loss combined with skin diseases. However, the mechanism by which different missense variants cause different hearing loss phenotypes is unknown. We have summarized 84 functionally studied missense variants and reviewed the clinical phenotypes and the molecular mechanisms that affected connexon and gap junction functions. We have also compiled a comprehensive dataset of *GJB2* missense variants from public databases and published literature and found that over 2/3 of the *GJB2* missense variants are currently classified as variants of uncertain significance. With the development of cutting-edge deep mutational scanning technology, we predict that the molecular mechanisms by which all missense variants of *GJB2* lead to different clinical phenotypes will be elucidated. These gained insights will facilitate the use of genetic testing in the prevention and control of hearing loss.

**Abstract:**

The *GJB2* gene is the most common gene responsible for hearing loss (HL) worldwide, and missense variants are the most abundant type. *GJB2* pathogenic missense variants cause nonsyndromic HL (autosomal recessive and dominant) and syndromic HL combined with skin diseases. However, the mechanism by which these different missense variants cause the different phenotypes is unknown. Over 2/3 of the *GJB2* missense variants have yet to be functionally studied and are currently classified as variants of uncertain significance (VUS). Based on these functionally determined missense variants, we reviewed the clinical phenotypes and investigated the molecular mechanisms that affected hemichannel and gap junction functions, including connexin biosynthesis, trafficking, oligomerization into connexons, permeability, and interactions between other coexpressed connexins. We predict that all possible *GJB2* missense variants will be described in the future by deep mutational scanning technology and optimizing computational models. Therefore, the mechanisms by which different missense variants cause different phenotypes will be fully elucidated.

## 1. Introduction

Hearing loss (HL) is a common sensory defect in humans. According to the World Health Organization, in 2021, more than 5% of the world’s population (~430 million) had debilitating HL and required rehabilitation [1]. There are an estimated 1–3 HL patients in China for every 1000 newborns [2,3]. HL affects speech acquisition and cognitive, social, and emotional development. Although various environmental factors can cause HL, most congenital deafness (50–60%) is attributable to genetic factors [4]. In general, hereditary HL is categorized as syndromic (30%) or nonsyndromic (70%), depending on the presence or absence of other clinical features. Currently, more than 120 genes are known to be causative genes for hereditary deafness (https://hereditaryhearingloss.org/, accessed on 18 November 2022), mainly encoding functional proteins involved in the development and function of the auditory system, such as transcription factors, structural proteins, ion channels, and gap junction proteins [5].

The *GJB2* gene is most commonly responsible for HL worldwide, accounting for up to 50% of nonsyndromic deafness. Different pathogenic variants predominate in different countries [6]. *GJB2* pathogenic variants include nonsense, small insertions or deletions, missense, and splicing variants. By 2021, the Human Gene Mutation Database (http://www.hgmd.cf.ac.uk/ac/index.php, accessed on 31 December 2021) had compiled 457 *GJB2* pathogenic variants, of which 303 (66.3%) were missense that covered ~70% (156/226) of the amino acid sites. Pathogenic missense variants in *GJB2* are a major cause of nonsyndromic HL and syndromic HL associated with skin disorders [7]. However, approximately 100 *GJB2* missense variants have definitive functions, and two-thirds have yet to be studied functionally. A systematic study of the effects on the protein functions of all of the *GJB2* missense variants will provide a basis for understanding the molecular mechanisms by which the *GJB2* missense variants lead to different clinical deafness phenotypes. Here we review the protein’s structural, positional, and functional changes and the clinical phenotypes caused by different missense variants in the *GJB2* gene. We mainly focus on those that have been functionally studied.

## 2. Structure and Function of the *GJB2* Gene

The *GJB2* gene is located on chromosome 13q12. It contains two exons and one intron. The coding region is located in exon 2 and encodes a protein containing 226 amino acids, also known as Connexin 26 (Cx26). Cx26 molecules share a common topology with other members of the connexin family, which includes two extracellular domains (E1, and E2), three intracellular domains (NT, CL, and CT), and four transmembrane domains (M1, M2, M3, and M4) (Figure 1A) [8]. Connexons (also known as hemichannels) are homomeric or heteromeric, depending on whether the same or different connexins are utilized as building blocks. They are trafficked to the cell surface in vesicles to connect the cytoplasm and the extracellular environment [9]. Two identical or different connexons can form a complete homotypic, heterotypic, or heteromeric gap junction (GJ) channel by docking through the extracellular domain (Figure 1B) [8,10,11,12]. GJs mediate intercellular communication by allowing the diffusion of ions, metabolites, and small signaling molecules, which are crucial for cell growth, differentiation, and development [13,14]. The human Cx26 gene is highly conserved and ablation of Cx26 in mice is embryonic lethal, indicating the essential functional roles of Cx26 in maintaining hemostasis of the human body [15]. Currently, there are several hypotheses for the pathogenic mechanism of Cx26-related hearing loss caused by defective hemichannel or gap junction channel function: potassium recycling disruption, adenosine-triphosphate-calcium signaling propagation disruption, and energy supply dysfunction [16,17,18].

## 3. Effect of *GJB2* Missense Variants on the Cx26 Protein

The life cycle of Cx26 includes protein translation, post-translational modifications, degradation of Cx26 molecules, assembly into hemichannel, intracellular trafficking of connexons to the plasma membrane, docking of the connexons from neighboring cells to form GJs, and interactions between other coexpressed connexins (Figure 2) [19,20,21]. The most common variant type for the *GJB2* gene is frameshift, such as c.35delG in Europeans and c.235delC in East Asians. These truncating variants usually lead to Cx26 lacking one or several of the transmembrane segments and intervening loops, which hampers Cx26 folding and oligomerization, resulting in retention at the endoplasmic reticulum (ER) and ultimately causing a total loss of function [22].

As the most abundant variant type for the *GJB2* gene, different missense variants can disrupt intercellular communication by affecting all of the Cx26 life cycle stages [20,22]. A recent review categorized *GJB2* missense variants based on whether the affected residues were involved in maintaining the structural stability or impaired the function of Cx26 [22]. In the present review, we summarize missense variants based on the functional consequence revealed by in vitro functional studies: impaired trafficking of connexons to the plasma membrane, reduced hemichannel or GJ functions, cell death caused by elevated hemichannel activity, and dominant or trans-dominant effects on wildtype Cx26 or other coexpressed connexins [23,24]. To date, roughly 100 *GJB2* missense variants with clear functions have been identified, and the pathogenesis of most missense variants remains unclear [22,25]. Therefore, it is crucial to investigate the effects of missense variants on protein functions and the molecular mechanisms by which different missense variants lead to different clinical phenotypes.

Cx26 is transcribed in the nucleus and trafficked to the endoplasmic reticulum (ER) for translation into proteins. Hydrolases in lysosomes are mainly used to degrade mistranslated proteins. Wildtype connexins oligomerize in the ER/Golgi. Microtubules assist hemichannels to traffic to the cell surface to allow communication between the cellular cytoplasm and the extracellular space. Hemichannels dock with the hemichannels of adjacent cells, forming gap junctions to traffic ions and other low-molecular-weight components between cells. ER: Endoplasmic reticulum

## 4. Association between *GJB2* Missense Variants with Clinical Phenotypes

Pathogenic variants in *GJB2* can cause nonsyndromic autosomal recessive or dominant HL [26] and syndromic HL associated with skin disease [7], including palmoplantar keratoderma (PPK) with deafness [27], Vohwinkel syndrome [28], keratitis-ichthyosis-deafness (KID) syndrome [29], hystrix-like ichthyosis with deafness (HID) [30], and Bart–Pumphrey syndrome (BPS) [31]. The skin abnormalities in these syndromic forms include follicular hyperkeratosis, palmoplantar stippled keratoderma, knuckle pads, leukonychia, and spotty hyperpigmentation [32]. The mechanisms leading to the different clinical phenotypes still need to be fully characterized. We have summarized these missense variants where there are functional studies and have grouped them into four general scenarios (Figure 3). These in vitro functional studies were performed using model cell lines, including transfected HeLa, HEK-293, NEB1 cells or the cochlear-relevant HEI-OC1 cells from mice, and the paired Xenopus oocytes expression system. The models were used to investigate the cellular localization and membrane trafficking of the mutant proteins, their ability to form functional gap junctions, and their relationship with the Cx26 wild type and other expressed connexins caused by missense variants [20].

### 4.1. Nonsyndromic HL Caused by GJB2 Missense Variants

#### 4.1.1. Nonsyndromic HL Caused by GJB2 Recessive Missense Variants

Recessive variants in the *GJB2* gene only cause nonsyndromic HL. To date, 50 autosomal recessive variants have been functionally studied. Some (16/50) have defective trafficking, some (7/50) have impaired trafficking, and others (24/50) have normal trafficking to the plasma membrane but have functionally impaired hemichannels and GJ channels (Table 1, see notes for the difference between defective trafficking and impaired trafficking). The recessive *GJB2* missense variants that have been functionally studied are distributed in all structural domains except the C-terminal domain. There are more variants in the N-terminal (6/50), EC1 (6/50), TM2 (8/50), CL (7/50), EC2 (7/50), and TM4 (11/50) structural domains, and only a small number of variants in the EC1 (3/50) and TM3 (2/50) structural domains.

Some missense variants largely accumulate in the endoplasmic reticulum (ER) or Golgi around the nucleus. They cannot translocate properly to the cell membrane to form GJ channels and affect protein function. In vitro evaluation of the G12V variant showed that the mutant protein was entirely intracellular, clustered in large perinuclear vesicles, and showed no evidence of membrane targeting [33]. The mutant protein of T86R did not form GJs because the mutant protein was retained in the cells [34]. However, when the mutant was coexpressed with wildtype Cx26, there was normal ionic and biochemical coupling, consistent with the recessive nature. Variants that share this pathogenic mechanism include M1V, R32C, R32H, I35S, W77R, S85Y, P173R, R184P, K188R, S199F, G200R, I203K, L205P, and T208P (Table 1).

**Table 1 biology-12-00505-t001:** Molecular mechanisms and phenotypes of nonsyndromic hearing loss variants.

Variants	Locations	Inheritance Patterns	Trafficking	Hemichannels	Gap Junction Channel	Clinical Phenotype	References
M1V	NT	AR	defect	/	/	severe/profound	[35,36]
T8M	NT	AR	normal	/	impaired	severe/profound	[37,38,39]
L10P	NT	AR	normal	impaired	impaired	profound	[39,40]
G12V	NT	AR	defect	/	/	mild/moderate	[33,41]
N14D	NT	AR	normal	impaired	impaired	moderate	[42]
S19T	NT	AR	impaired	/	defect	profound	[33]
R32C	TM1	AR	defect	/	/	/	[43]
R32H	TM1	AR	defect	/	/	severe/profound	[44,45]
I33T	TM1	AR	normal	/	defect	severe/profound	[46]
M34T	TM1	AD	normal	impaired	impaired	mild/moderate/severe	[35,41]
I35S	TM1	AR	defect	/	/	severe/profound	[46]
V37I	TM1	AR	normal	defect	impaired	mild	[41,47,48]
A40G	TM1	AR	normal	defect	defect	profound	[48,49]
W44C	EC1	AD	normal	/	Impaired	severe/profound	[50,51]
W44S	EC1	AD	impaired	impaired	impaired	/	[52,53]
G45R	EC1	AD	normal	impaired	impaired	mild	[54]
D46E	EC1	AD	normal	impaired	defect	moderate	[34]
E47K	EC1	AR	normal	impaired	defect	/	[55,56]
E47Q	EC1	AR	normal	impaired	defect	/	[56]
T55N	EC1	AD(YES)	defect	/	/	severe/profound	[57]
G59V	EC1	AD	normal	/	defect	profound	[58,59]
I71N	EC1	AR	impaired	impaired	/	severe/profound	[60]
W77R	TM2	AR	defect	/	/	severe/profound	[38,50,61]
I82M	TM2	AR	normal	defect	/	profound	[38,59]
V84L	TM2	AR	normal	normal	impaired	profound	[41,62,63]
S85Y	TM2	AR	defect	/	/	/	[56]
T86R	TM2	AR	defect	/	/	profound	[34]
A88S	TM2	AR	normal	normal	impaired	profound	[63,64]
L90P	TM2	AR	impaired	impaired	/	severe/profound	[33,35,38]
V95M	TM2	AR	normal	normal	impaired	profound	[49,62,63]
H100L	CL	AR	normal	impaired	impaired	profound	[49,56]
H100Y	CL	AR	normal	impaired	impaired	/	[56]
G109V	CL	AR	normal	impaired	defect	profound	[39,65]
S113R	CL	AR	/	/	defect		[47]
R127H	CL	AR	normal	impaired	impaired	severe/profound	[33,45,56]
R127L	CL	AR	normal	impaired	impaired	/	[56]
R143Q	TM3	AD	normal	/	defect	profound	[53,66]
R143W	TM3	AR	/	/	defect	profound	[37,41]
V153I	TM3	AR	/	/	defect	severe/profound	[37,38]
F161S	EC2	AR	impaired	/	impaired	/	[35]
M163L	EC2	AD(YES)	defect	/	/	mild/moderate	[67]
M163V	EC2	AD	normal	/	impaired	severe/profound	[24,38]
W172C	EC2	AR	impaired	impaired	/	profound	[20,68]
W172R	EC2	AR	normal	/	impaired	/	[46]
P173R	EC2	AR	defect	/	/	severe/profound	[35,69]
D179N	EC2	AR	normal	/	impaired	severe	[53,70]
R184Q	EC2	AD	normal	/	defect	profound	[51,53]
R184P	EC2	AR	defect	/	/	severe/profound	[35,44,49]
K188R	EC2	AR	defect	/	/	/	[71]
M195V	TM4	AD(YES)	defect	/	/	severe/profound	[56,72]
M195T	TM4	AR	normal	/	impaired		[71]
M195L	TM4	AR	normal	impaired	impaired	/	[56]
A197S	TM4	AD	normal	/	impaired	profound	[71,73]
S199F	TM4	AR	defect	/	/	severe/profound	[71,74]
G200R	TM4	AR	defect	/	/	severe	[71,75]
C202F	TM4	AD	normal	impaired	impaired	mild/moderate	[71,76]
I203T	TM4	AR	normal	impaired	impaired	profound	[71,77]
I203K	TM4	AR	defect	/	/	profound	[71,73]
L205V	TM4	AR	normal	impaired	impaired	profound	[71,78]
L205P	TM4	AR	defect	/	/	moderate/severe/profound	[71,79]
N206S	TM4	AR	normal	impaired	impaired	mild/moderate/severe	[71,77]
T208P	TM4	AR	defect	/	/	/	[71]
L214P	TM4	AR	/	/	defect	profound	[37,73]

NT: N-term, TM: transmembrane, EC: extracellular loop, CL: cytoplasmic loop, AD: autosomal dominant, AR: autosomal recessive. Defective trafficking: the mutant is completely retained within the cell and cannot traffic to the cell membrane. Impaired trafficking: only a small fraction of the mutant is able to traffic to the cell membrane. The WHO hearing loss classification standard (1997): normal (≤25 dB), mild (26–40 dB), moderate (41–60 dB), severe (61–80 dB), and profound (≥81 dB).

Some mutants can partially traffic to the cell membrane surface, but those that reach the cell membrane are insufficient to form functional GJ channels or functionally impaired GJ channels. The S19T and L90P mutants showed intracellular and cell membrane localization, but the level of GJ plaque formed was low [33]. F161S showed weak membrane localization in immunofluorescence studies, confirming that few of these mutant proteins were transferred to the cytosol [35]. Similarly, the W172C mutant is translocated chiefly to the cell membrane. However, the function is significantly impaired compared to the wildtype, as revealed by propidium iodide (PI) dye loading assay, showing the significantly reduced permeability of the mutated hemichannels [20].

Other mutants can traffic to the cell membrane but form GJ channels with no or impaired functions. The V37I and A40G mutants have similar plasma membrane localization to wildtype Cx26 [48]. However, both mutants were less efficient in forming GJ patches than the wildtype. Moreover, none formed functional hemichannels at low extracellular calcium. Kim et al. evaluated seven of ten Cx26 mutants that could translocate to the plasma membrane to form GJ plaques and found that E47K, E47Q, H100L, H100Y, and R127L did not function normally as homo-oligomeric GJ channels [56]. However, all seven variants could function normally as hetero-oligomeric GJ channels. When the mutants formed a heteromeric connexon with wildtype Cx26, the ionic coupling of the GJ channels did not significantly differ from that of the GJs composed of wildtype Cx26 alone.

In conclusion, trafficking defects, abnormal hemichannel activity, and abnormal GJ channel function are the main pathogenic mechanisms of recessive pathogenic variants. However, most *GJB2* missense variants have not been functionally studied, or the functional findings are controversial. Therefore, further studies are needed to determine the underlying molecular mechanisms and guide clinical diagnosis and treatment.

#### 4.1.2. Nonsyndromic HL Caused by GJB2 Dominant Variants

Compared to recessive Cx26 variants, few dominant pathogenic variants have been identified that cause nonsyndromic HL. These dominant variants only alter the function of the mutant protein; they do not affect wildtype or other coexpressed proteins and therefore only cause nonsyndromic HL. There are 14 *GJB2* nonsyndromic dominant variants in the TM1 (1/14), EC1 (6/14), TM3 (1/14), EC2 (3/14), and TM4 (3/14) structural domains. They include three variants showing trafficking defects, one showing impaired trafficking, and ten showing normal trafficking to the cell membrane (Table 1).

The nonsyndromic dominant variants include three mutants with trafficking defects. In cells transfected with the T55N-EYFP mutant, EYFP localized intracellularly, indicating that the mutation impaired protein trafficking to the plasma membrane [57]. However, the mutant protein M163L was not trafficked to the plasma membrane but did traffic to the cell membrane, where it co-localized with the wildtype Cxs when coexpressed with wtCx26 or wtCx30 [67]. Cells expressing M163L alone acquired a rounded and detached morphology, suggesting the mutant was associated with increased cell death. Another dominant mutant, M195L, remained in the cytoplasm, particularly in the ER, and could not traffic to the cell membrane [56].

Only one nonsyndromic dominant variant, W44S, has been functionally investigated and has impaired trafficking. W44S resulted in a protein localized to the membrane and dispersed intracellularly [52]. However, dye transfer studies have revealed a significant decrease in the ability of the W44S mutant to transfer neurobiotin to neighboring cells, which implies a compromised ability of the mutant protein to form normal functional channels. Usually, variants that cause nonsyndromic HL do not adversely affect wildtype proteins. However, W44S produces proteins with a dominant negative effect on Cx26 and Cx30 [52]. Sabrina et al. obtained the same results where mutants coexpressed with Cx26 and Cx30 did not transfer Lucifer Yellow dye [53].

Nine variants have normal trafficking capacities to the cell membrane. The M34T mutant has strong membrane localization, but the hemichannel assembly is disrupted, and a neurobiotin injection showed decreased tracer coupling in M34T transfectants [35]. The G46E mutant exhibits similar cell membrane localization to the wildtype Cx26 and can co-assemble with it into GJ channels but with reduced hemichannel activity and biochemical coupling [34]. Other dominant variants with similar pathogenic mechanisms are W44C, G45R, G59V, R143Q, R184Q, A197S, and C202F.

Dominant variants causing nonsyndromic HL do not induce cell death. However, in an in vitro study, the M163L mutant protein targeted the plasma membrane with a trafficking defect and was associated with increased cell death [67]. M163L is the only pathogenic variant identified in nonsyndromic HL that causes cell death. Therefore, the increased cell death is a pathogenic mechanism leading to nonsyndromic HL that needs further study.

#### 4.1.3. Relationship between Variants Causing Nonsyndromic HL and Clinical Phenotype

In clinical practice, *GJB2* pathogenic missense variants can lead to varying degrees of HL. Nonsyndromic HL due to *GJB2* recessive missense variants is primarily congenital, bilaterally symmetric, and non-progressive, with severe-to-profound severity. However, some variants cause mild to moderate HL. Recessive variants might lead to milder phenotypes due to their compound heterozygous partners. Compound heterozygosity with truncating variants may lead to more severe phenotypes, whereas missense variants that do not have serious consequences may lead to less severe HL [80]. Dominant variants cause mainly postlingual progressive hearing loss. Dominant variants, such as G45R, may cause milder HL because the mutant can traffic to the cell membrane with only partially impaired hemichannel or GJC function [54]. On the other hand, the M163L variant cannot traffic itself, but when co-expressed with the wildtype Cx26, it can traffic to the cell membrane, thus leading to a milder phenotype [67].

V37I can cause mild to moderate sensorineural HL. A review of the synthesized information of the ClinGen Hearing Loss Expert Panel identified significant overexpression of the V37I variant in HL patients compared to the normal population. The panel concluded that the V37I variant in *GJB2* is pathogenic for autosomal recessive nonsyndromic HL with variable expressivity and incomplete penetrance [81]. The biallelic p.V37I variant is associated with steadily progressive HL with increasing incidence over time, and most biallelic p.V37I individuals may develop significant HL in later adulthood [82].

### 4.2. Syndromic HL Caused by GJB2 Missense Variants

#### 4.2.1. Syndromic HL Caused by GJB2 Dominant Variants

More than 20 *GJB2* missense variants cause syndromic HL, most of which are dominantly inherited. Functional studies have found that dominant syndromic variants are not present in the C-terminal and EC2 structural domains but are clustered in the N-terminal (5/21) and EC1 (9/21) structural domains, and in the TM1 (2/21), TM2 (3/21), TM3 (1/21), and TM4 (1/21) structural domains, respectively. Variants causing syndromic phenotypes are usually located at conserved amino acid sites and cause more severe phenotypes than dominant variants causing nonsyndromic HL. Moreover, the mechanisms causing syndromic HL are not identical to those causing nonsyndromic HL. Besides trafficking defects, abnormal hemichannel, and GJ channel functions, most dominant variants induce cell death or have dominant-negative or trans-dominant effects on wildtype and other Cxs, resulting in HL combined with a skin disease (Table 2).

Compared to pathogenic variants causing nonsyndromic HL, pathogenic variants that cause syndromic HL usually lead to increased hemichannel activity. However, some mutant proteins have impaired hemichannel activity; for example, the S17F variant resulted in a complete loss of hemichannel activity [85]. Dye uptake assays showed that cells expressing the N54K mutant demonstrated a highly variable incidence of PI uptake, and cells expressing the S183F mutant displayed no increase in PI uptake, indicating nonfunctional hemichannels [24]. Marziano et al. [52] and Albuloushi et al. [87] obtained similar results for the hemichannel activity of the D66H and F142L mutants. Both variants resulted in syndromic HL and had reduced hemichannel activity.

A common feature of syndromic HL is increased induction of cell death. Six known variants induce increased cell death. In experiments, 53% of cells transfected with the G11E mutant died after 24 h; however, only 13% of cells transfected with the wildtype died [83]. Cells expressing G12R, N14K, and D50N experienced increased cell death that correlated with larger hemichannel currents than the wild-type-expressing cells [85]. Gerido et al. [91] proved that Cx26-G45E hemichannels displayed significantly greater whole-cell currents than wildtype Cx26, leading to cell lysis and death. A40V produced abnormal hemichannel activity when expressed in *Xenopus* oocytes, eventually leading to cell lysis and death [92].

Another distinctive feature of variants causing syndromic HL is a dominant-negative or trans-dominant effect on wildtype Cx26 or other Cxs. For example, HeLa cells cotransfected with G45R and wildtype Cx26 could form GJs, whereas G45E (causing nonsyndromic HL) and wildtype Cx26 did not [54]. The N54K mutant protein was retained intracellularly and displayed a dominant or transdominant effect on wildtype Cx26 and coexpressed Cx30 and Cx43. The S183F mutant formed some GJ plaques but was primarily retained within the cell and exhibited only a mild trans-dominant effect on coexpressed Cx30 [24]. Coexpression of the S183F and H73R mutants with wildtype Cx43 showed trans-dominant inhibition of Cx43 GJs, without affecting Cx43 protein synthesis [95]. R75W had a dominant negative effect on wildtype Cx26 [46] and a trans-dominant effect on wildtype Cx30 [52]; however, R75Q only had a trans- dominant effect on wildtype Cx30 [53]. Similarly, the dominant actions of the G59A and D66H mutants were only on Cx30 and Cx26, respectively [52].

Other dominant pathogenic variants that cause syndromic HL, such as I30N, D50A, and D50Y, have only been explored regarding whether they can traffic properly and have hemichannel functions. It is unclear whether the pathogenic variants affect cell death, or wildtype and other connexins proteins. Therefore, further studies are needed to elucidate clear pathogenesis.

#### 4.2.2. Relationship between Pathogenic Variants Causing Syndromic HL and Clinical Phenotypes

Syndromic variants in Cx26 are associated with various skin disorders and always present with autosomal dominant inheritance. The features of syndromic HL usually include sensorineural HL and various epidermal abnormalities such as palmoplantar keratosis (thickened skin on the palms and soles of the feet), knuckle pads, finger (toe) nail abnormalities, ichthyosis, and false hoop toe/toe breaks [32].

Pathogenic variants associated with cell death and dominant-negative and trans-dominant effects, such as G11E, G45E, and H73R, usually lead to more severe phenotypes such as KID syndrome or PPK. Pathogenic variants without these features generally cause only the milder phenotype of Vohwinkel syndrome. However, several pathogenic variants, including N14K, I30N, and D50A, did not induce cell death or have dominant negative or trans-dominant effects on wildtype and other connexins. However, they still caused KID syndrome, suggesting that these variants may have other pathogenic mechanisms that require further exploration (Table 2).

## 5. The Gap between Possible and Functionally Studied Missense Variants of the *GJB2* Gene

With the development of high-throughput sequencing technology, genetic testing strategies, such as target gene sequencing and whole-exome sequencing, are widely used to diagnose the causes of HL. These will promote the prevention and treatment outcomes of genetic deafness. In addition, joint newborn hearing and genetic screening for HL can significantly improve the benefits of hearing screening in China [2,3]. Medical geneticists classify genetic variants found in genetic testing into five tiers, including pathogenic, likely pathogenic, variants of uncertain significance (VUS), likely benign, or benign based on the ACMG guidelines [98]. On the other hand, the advancement of genetic testing technologies also results in more VUSs due to the increased number of genes being analyzed, which hinders HL prevention and control.

The *GJB2* gene is the most prevalent gene responsible for HL worldwide. Although less common than truncating variants, missense variants are the most abundant variant type of the *GJB2* gene [99]. Cx26 contains 226 amino acid residues, and each residue can be mutated to one of the other 19 amino acids; therefore, there are 4294 possible missense variants. Of these 4294 variants, 1334 can be achieved by a single base pair variant in a codon (simple missense variants), while 2960 can only be achieved by variant in more than two bases in a codon (complex variants) (Figure 4). We have compiled a comprehensive dataset of *GJB2* missense variants based on public databases, including HGMD, Clinvar, and the Deafness Variation Database, and literature articles [99]. We have classified these 1334 simple missense variants into five tiers (Appendix A), and approximately 84.2% of missense variants are VUS (Figure 4). Information on the effects of all VUSs on Cx26 is required to guide clinical diagnosis, treatment, and prevention.

Simple missense variants were defined as missense variants that can be achieved by a single base pair variant in a codon, while complex ones can only be achieved by a variant in more than two bases in a codon. Each column indicates one amino acid (AA) residue in Cx26 (upper panel: 1–113, lower panel: 114–226). The 20 rows represent all 20 possible AA substitutions. WT, wildtype: white; Com, complex missense variants: grey; B: Benign; LB: likely Benign; VUS: Variants of Uncertain Significance; LP: Likely Pathogenic; P: Pathogenic.

It is impossible to rapidly determine the functions of the many possible *GJB2* VUSs, because the traditional experimental methods of overexpressing mutants in cell lines waste manpower and resources. An alternative solution is to predict pathogenicity using computational methods, which can predict the effects of the missense variants on a large scale. However, based on sequence homology, protein structure, and evolutionary conservation, these computational methods have their merits and limitations, and optimal tool selection is not always possible [100].

Another high-throughput method to solve VUSs is deep mutational scanning (DMS) [101], which is possible due to the development of synthetic DNA technology and the decreasing sequencing costs. DMS technology has been widely used to study the function of variants in clinically actionable genes, such as *BRCA1* [102], *PPARG* [103], *TP53* [104], *NUDT15* [105], *MSH2* [106], and *EGFR* [107]. DMS includes three parts: the construction of a variant library; the identification and validation of the selection system for the protein function of interest and the introduction of the variant library into the selection system and subjected to selection; and the calculation of a functional score for each variant based on the change in the frequency of the variant during the selection [108] (Figure 5). Among the three parts, selection system, i.e., functional assays, needs to be properly designed and validated. Several commonly used functional assays are widely applied in DMS, such as cytotoxicity assays implemented in studying *EGFR* [107] and *SCN5A* variants [109], protein affinity assay in studying functional interaction sites in *CXCR4* and *CCR5* [110], reporter assay in *ADRB2* study [111], and resting membrane potential assay for a K^+^ channel *KCNJ2* [112]. For the *GJB2* gene, a library of all possible 4294 missense variants will be created and subjected to the selection system. The selection system will mainly focus on the trafficking of connexin, hemichannel activity, GJC function, cell death, and dominant-negative or trans-dominant effects on coexpressed Cxs. Applying DMS to the *GJB2* gene will elucidate the molecular mechanisms by which different *GJB2* missense variants cause different clinical HL phenotypes and promote the application of genetic testing in HL prevention and control.

## 6. Conclusions

*GJB2* is one of the most important genes responsible for hearing loss, and missense variants are the most abundant variant type. Pathogenic missense variants can lead to both nonsyndromic and syndromic hearing loss combined with skin diseases; however, the molecular mechanisms leading to the different clinical phenotypes are not fully understood. We have summarized the 84 missense variants that have been functionally studied and investigated the molecular mechanisms that affected hemichannel and gap junction functions, including connexin biosynthesis, trafficking, oligomerization into connexons, permeability, and interactions between other coexpressed connexins. We have also compiled a comprehensive dataset of *GJB2* missense variants from public databases and published literature and found that most missense variants have not been functionally studied and have been rated as variants of unknown significance. Therefore, with the recently developed deep mutational scanning technology, which draws on high-throughput DNA sequencing to assess the functional capacity of a large number of variants of a protein simultaneously [101], we hope to elucidate the functional consequence for all possible *GJB2* missense variants. These results will help to reveal the mechanisms by which different missense variants cause different phenotypes and to provide guidance for genetic counseling and deafness prevention and control.

## Figures and Tables

**Figure 1 biology-12-00505-f001:**
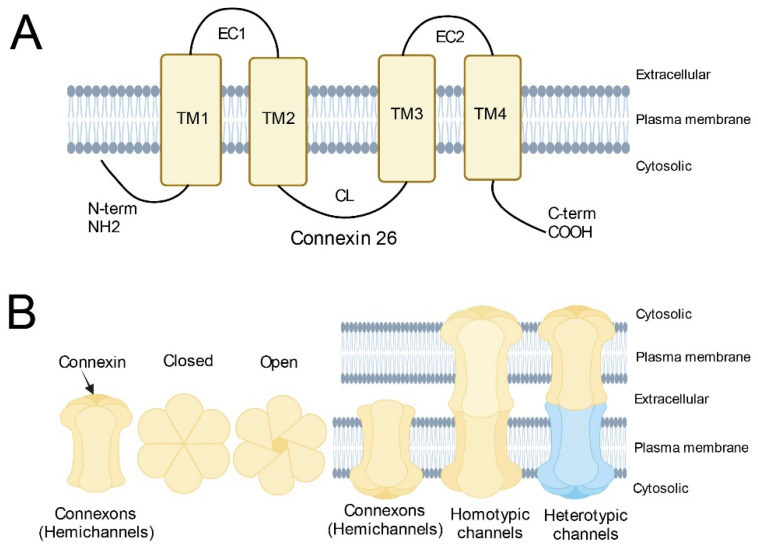
Schematic diagram of Connexins. (**A**) Basic structure of Connexin 26; (**B**) Structure of connexons and (homotypic/heterotypic) gap junction channels. Homotypic gap junction channels are composed of two identical connexons. Heterotypic gap junction channels are composed of two different connexons.

**Figure 2 biology-12-00505-f002:**
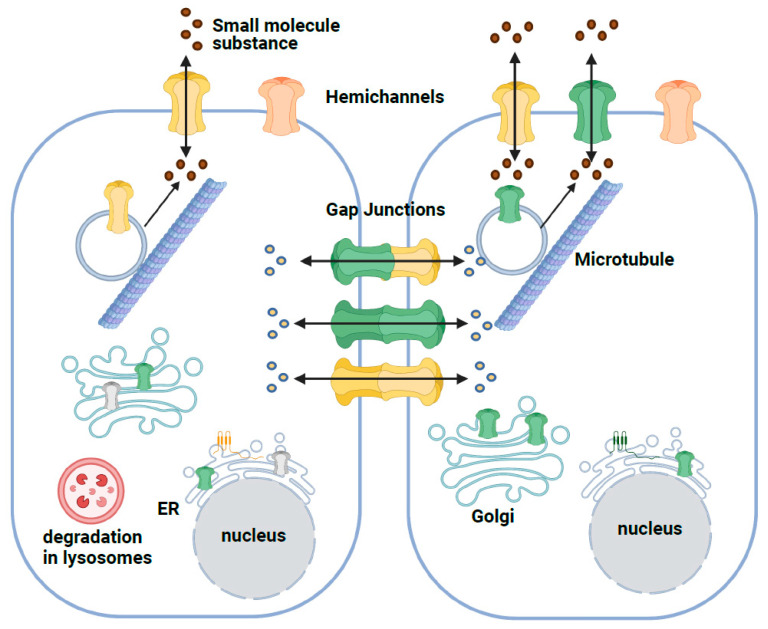
Life cycle of Cx26. Green hemichannels represent Cx26; the hemichannels in other colors represent other connexins. Green gap junction channels represent homotypic gap junction channels composed of Cx26. Yellow gap junction channels represent homotypic gap junction channels composed of other connexins. The green and yellow gap junction channels are heterotypic or heteromeric or other types of junction channels composed of Cx26 and other connexins.

**Figure 3 biology-12-00505-f003:**
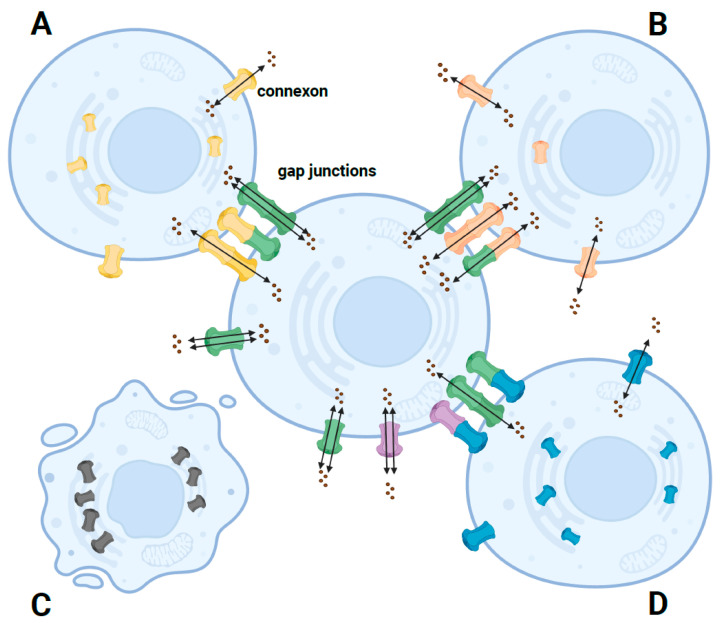
Schematic diagram of the pathogenic mechanism of different phenotypes caused by *GJB2* missense mutants. (A) In autosomal recessive nonsyndromic HL, most missense mutants are retained in the cell. A small fraction can traffic to the cell membrane, forming nonfunctional or functionally impaired hemichannels or functional homozygous GJ channels but not functional heterozygous channels. (B) In autosomal dominant nonsyndromic HL, most missense mutants are correctly trafficked to the cell membrane but form impaired GJ channels. (C) In autosomal dominant syndromic HL, some missense mutants can induce cell death due to increased calcium channel activity, resulting in increased hemichannel currents. (D) In autosomal dominant syndromic HL, most missense mutants have dominant-negative or trans-dominant effects on wildtype or other proteins, so the connexons coexpressed with the mutant are suppressed and form nonfunctional GJ channels. Green connexons represent the wildtype Cx26; yellow, orange, and blue connexons represent mutated Cx26; purple connexons represent other connexons.

**Figure 4 biology-12-00505-f004:**
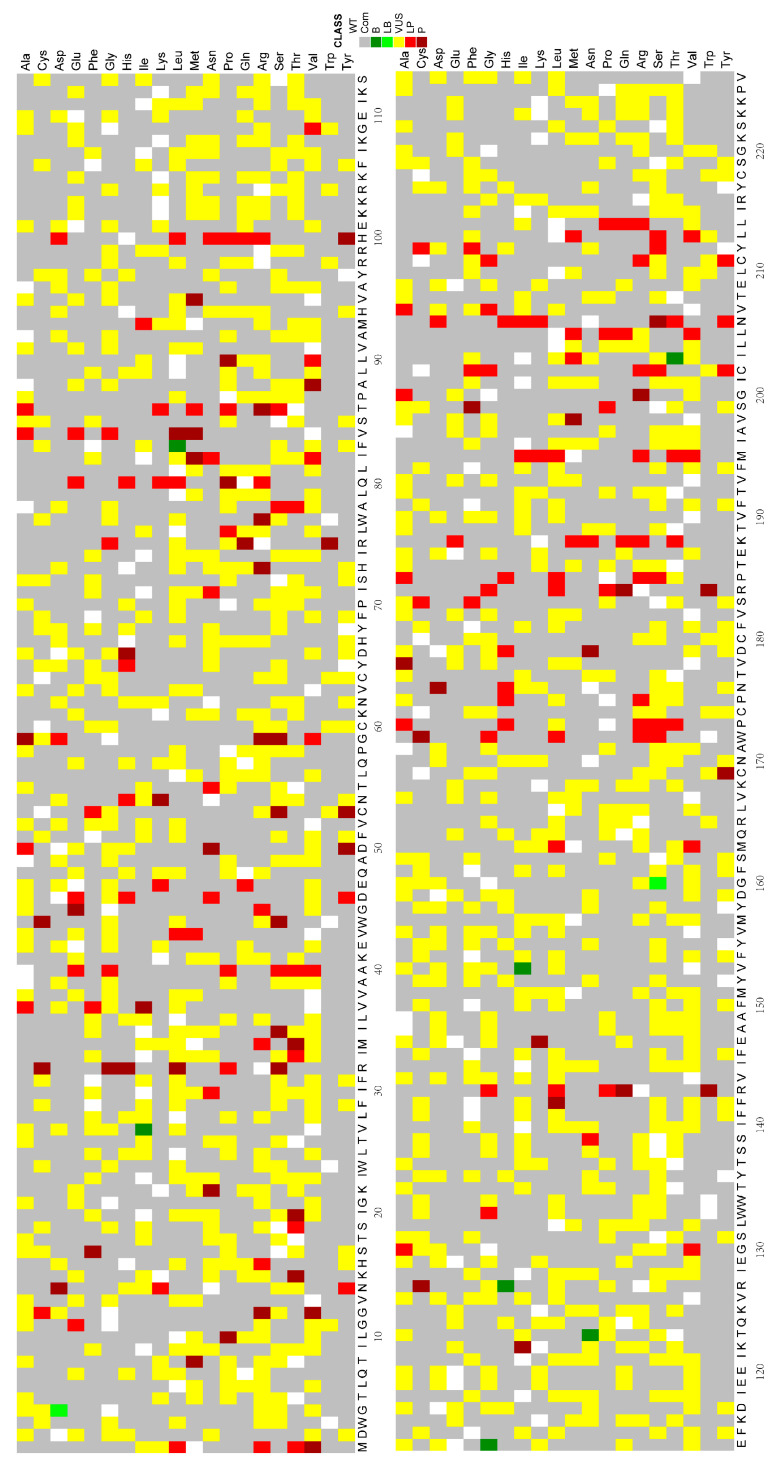
The type (simple and complex) and pathogenicity classification of *GJB2* missense variants.

**Figure 5 biology-12-00505-f005:**
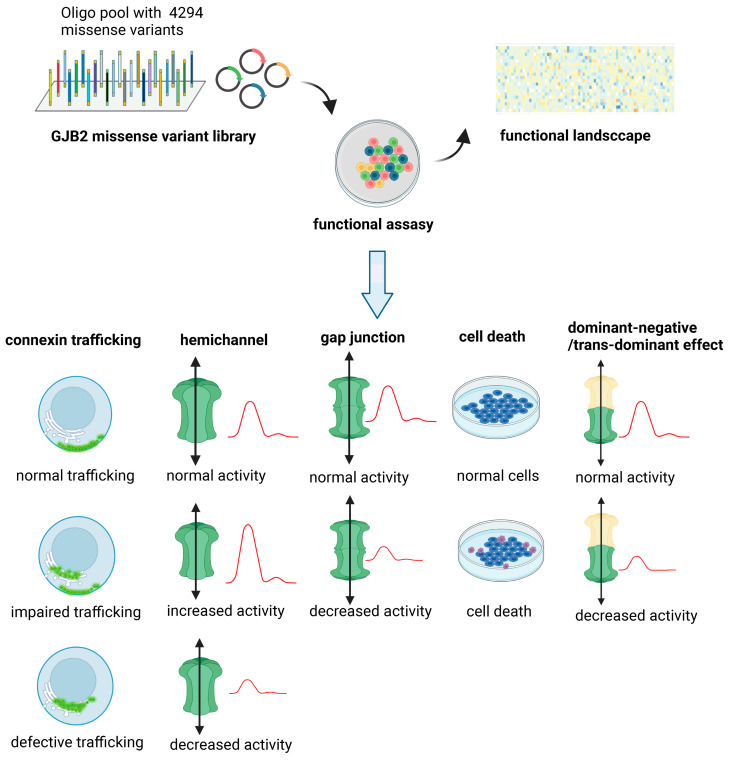
Schematic diagram of deep mutational scanning on the *GJB2* missense variants. First, a variant library containing 4294 missense variants will be constructed. Second, functional assays will be performed to evaluate the variants’ effects on connexin trafficking, hemichannel activity, GJC function, cell death, and coexpressed Cxs. Finally, a functional profile will be obtained based on the change in the frequency of the variants during the selection. Green connexons represent mutated Cx26; the yellow connexons represent wildtype Cx26 (dominant-negative effect assay) or other wildtype connexons (trans-dominant effect assay).

**Table 2 biology-12-00505-t002:** Molecular mechanisms and phenotypes of syndromic hearing loss variants.

Variants	Locations	Inheritance Patterns	Trafficking	Hemichannels	Gap Junction Channel	Cell Death	Dominant Negtive/Trans-Dominant Effects	Clinical Phenotype	References
G11E	NT	AD	defect	Increased	/	Increased	/	KID	[83,84]
G12R	NT	AD	impaired	Increased	impaired	Increased	YES	KID	[29,85,86,87]
N14Y	NT	AD	impaired	Increased	impaired	/	/	KID	[86,88]
N14K	NT	AD	defect	Increased	impaired	Increased	/	KID	[24,85]
S17F	NT	AD	impaired	defect	impaired	/	/	KID	[29,85,86]
I30N	TM1	AD	impaired	Increased	/	/	/	KID	[89,90]
A40V	TM1	AD	/	Increased	impaired	Increased	/	KID	[91,92]
G45E	EC1	AD	impaired	Increased	impaired	Increased	YES	KID	[54,91]
D50A	EC1	AD	impaired	Increased	/			KID	[93]
D50N	EC1	AD	defect	Increased	impaired	Increased	/	KID, HID	[30,83,85]
D50Y	EC1	AD	impaired	Increased	/	/	/	KID	[89]
N54K	EC1	AD	defect	impaired	impaired	/	YES	BPS, PPK + deafness	[24,31,44]
G59A	EC1	AD	defect	/	impaired	/	YES	PPK + deafness	[52,53]
Y65H	EC1	AD	impaired	/	impaired	/	/	Vohwinkel	[94]
D66H	EC1	AD	defect	impaired	impaired	/	YES/	Vohwinkel	[52,87]
H73R	EC1	AD	impaired	/	impaired	/	YES	PPK + deafness	[95,96]
R75Q	TM2	AD	normal	/	impaired	/	YES	PPK + deafness	[53]
R75W	TM2	AD	normal	/	impaired	/	YES	PPK + deafness	[52,53]
A88V	TM2	AD	impaired	Increased	/	/	/	KID	[93]
F142L	TM3	AD	impaired	impaired	impaired	/	YES/	Keratoderma-Deafness-Mucocutaneous Syndrome	[87,97]
S183F	TM4	AD	impaired	impaired	impaired	/	YES	PPK + deafness	[24]

NT: N-term, TM: transmembrane, EC: extracellular loop, CL: cytoplasmic loop, AD: autosomal dominant, AR: autosomal recessive.

## Data Availability

Not applicable.

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
