# Peer review of "Molecular Mechanisms and Clinical Phenotypes of GJB2 Missense Variants"

_biology, 2023, doi:10.3390/biology12040505_

Round 1
Reviewer 1 Report
Review
to manuscript “Molecular mechanisms and clinical phenotypes of GJB2 missense variants”
This review discussed the molecular mechanisms and clinical phenotypes of GJB2 missense variants. Based on these functionally determined missense variants, authors reviewed the clinical phenotypes and investigated the molecular mechanisms that affected hemichannel and gap junction functions, including connexin biosynthesis, transportation, oligomerization into connexons, permeability and interactions between other co-expressed connexins.
Minor comments:
1) Nomenclature of genes needs for correction:
- The names of human genes should be written in capital letters, in italics;
- Non-human: in capital letters, in italic;
2) All figures need more explanation of their individual elements in figure captions. For example, figure 1, from this figure it's not very clear what it means homotypic/heterotypic connexins channels? It’s Cx26 or other connexins?
3) The term “mutation” is not clear. Accordingly by the ACMG guidelines for description of the pathogenic effect of the different variants we must be applied terms classified at 5 categories (“pathogenic”, “likely pathogenic”, “uncertain significant”, “likely benign”, “benign”).
Recommendation
Accept after minor comments
Reviewer 2 Report
Title should include sensorineural hearing loss.
It might be necessary to explain the difference of nonsense and missense mutations.
Table 1: What is the difference of severe and profound hearing losses? Maybe definition of. hearing loss classification should be included.
Figure 3: Legend including the explanation of A~D in required.
Before discussing the pathophysiology of missense mutation, path-mechanism of hearing loss caused by general Gjb2 variants should be included. In addition discussing the hearing loss theories by hemichannel and gap junction might help readers' understanding.
Figure 2 legend should be different font: "Cx26~reticulum" and figure 2 title mentioned consequences of missense variants which is not described neither in figure and legend.
Please explain dominant or trans-dominant effects.
Please explain the experimental methods of in vitro functional studies, are the sam methods are used for all the studies in Table 1.
Please check the red highlighted table 2 in page 9.
Please explain the possible theory of increased hemichannel activity induced cellular death.
Most importantly, it is very crucial to acknowledge the nature of hearing loss. Authors mentioned the degree of hearing loss by classifying them to groups. However authors should mention whether hearing loss is congenital(prelingual) or progressive(pre or post lingual). This difference could be related to possible pathomeachanism of hearing loss caused by Cx dysfunctions.
As authors mentioned clarifying the VUS to enhance treatment, I think authors have to clearly discuss about therapeutic option for missense mutation, gene therapy or Crisper-Cas9 correction. Would it be different from the therapeutic option for nonsense mutation.
Currently it is relatively difficult to understand the manuscript. Possibly several review processes would be necessary to increase the structure and organization of manuscript.
Reviewer 3 Report
Dear Authors,
Congratulation to your great and significant work.
The paper: “Molecular mechanisms and clinical phenotypes of GJB2 missense variants«. First, they described nicely the role of Cx26 and then they constructed a variant library containing 4,294 missense variants. Later on, they conducted functional assays to evaluate the variants’ effects on connexin transport, channel activity, GJC function, cell death, and coexpressed Cxs. As a result, they obtained a 111 functional profile based on the change in the frequency of the variant during the selection.
The data are in Table’s, but I would like you to add legend for each Table.
Other vice no comment.
Reviewer 4 Report
The submitted manuscript nicely present an overview of the pathogenicity of functionally characterized GJB2 variants reported in the recent literature, with the aim to define a general principle to categorize any given GJB2 variant that may arise.
The suggested approach to define the pathogenicity of GJB2 variants is for sure interesting and it would allow for a better understanding of the structure-function relationship of the protein as a whole but, as reported in the publication you refer to (52, Fowler et al, 2014) it can be a daunting enterprise, especially when dealing with membrane proteins in mammalian cells. I think the proposed approach would be more convincing with the support of a few practical examples of applicable selection methods on a high throughput scale from the recent literature.
In the following are a few adjustment I would suggest for the text:
"Traffic" or "trafficking" instead of "transport" (e.g.: line 129 pag. 4, tables); in my opinion it might be mistaken for the transport activity of a protein.
Clearly define the difference between "defective" and "impaired" (text and tables).
Add an appropriate reference at line 65, pag. 12.
Implement the information in all the figure legends, in order to improve the readability and comprehension of the figure itself in the context of the referring text (e.g.: Figure 3, clearly define A,B, C and D).
If possible, enlarge the fonts in Figure 4 to improve readability. Also, if possible, report the substituted aminoacid on the Y axis instead of letters in alphabetical order. Lastly, I think it would be helpful if the aminoacids on the X axis were numbered according to their position in the polypeptide.
What is the "relevant enzyme" at line 96 pag.3 ?
It is not clear to me whether the suggested approach has already been implemented in your group or if it is a suggestion for further research, especially the very last paragraph, after figure 5 (Lines 109-112, pag 13). In the former case, in my opinion, it would be outside the scope of a literature review. Please clarify the issue.
Finally, I think the supplementary materials are missing from the submission, or at least, the only supplemental file I could find is the list of authors and contributions.
Round 2
Reviewer 2 Report
I do not have further comment.
Author Response
Thanks.
Reviewer 4 Report
Dear Authors,
thank you for properly addressing all the issues from the first round of revision. I think the manuscript is significantly improved and I am pleased to inform you that from my side the it is ripe for publication, without any further modifications.
Author Response
Thanks.